# Facile Preparation of Mechanically Robust and Functional Silica/Cellulose Nanofiber Gels Reinforced with Soluble Polysaccharides

**DOI:** 10.3390/nano12060895

**Published:** 2022-03-08

**Authors:** Marco Beaumont, Elisabeth Jahn, Andreas Mautner, Stefan Veigel, Stefan Böhmdorfer, Antje Potthast, Wolfgang Gindl-Altmutter, Thomas Rosenau

**Affiliations:** 1Department of Chemistry, Institute of Chemistry of Renewable Resources, University of Natural Resources and Life Sciences Vienna, Konrad-Lorenz-Straße 24, 3430 Tulln, Austria; elisabeth.jahn@students.boku.ac.at (E.J.); stefan.boehmdorfer@boku.ac.at (S.B.); antje.potthast@boku.ac.at (A.P.); 2Faculty of Chemistry, Institute of Materials Chemistry and Research, Polymer and Composite Engineering (PaCE) Group, University of Vienna, Währinger Street 42, 1090 Vienna, Austria; andreas.mautner@univie.ac.at; 3Department of Material Sciences and Process Engineering, Institute of Wood Technology and Renewable Materials, University of Natural Resources and Life Sciences Vienna, Konrad-Lorenz-Straße 24, 3430 Tulln, Austria; stefan.veigel@boku.ac.at (S.V.); wolfgang.gindl-altmutter@boku.ac.at (W.G.-A.); 4Johan Gadolin Process Chemistry Centre, Åbo Akademi University, Porthansgatan 3, FI-20500 Turku, Finland

**Keywords:** nanocellulose, functional nanocomposite, aqueous process, sol–gel, hydrogels, aerogels, freeze-drying, cryogels

## Abstract

Nanoporous silica gels feature extremely large specific surface areas and high porosities and are ideal candidates for adsorption-related processes, although they are commonly rather fragile. To overcome this obstacle, we developed a novel, completely solvent-free process to prepare mechanically robust CNF-reinforced silica nanocomposites via the incorporation of methylcellulose and starch. Significantly, the addition of starch was very promising and substantially increased the compressive strength while preserving the specific surface area of the gels. Moreover, different silanes were added to the sol/gel process to introduce in situ functionality to the CNF/silica hydrogels. Thereby, CNF/silica hydrogels bearing carboxyl groups and thiol groups were produced and tested as adsorber materials for heavy metals and dyes. The developed solvent-free sol/gel process yielded shapable 3D CNF/silica hydrogels with high mechanical strength; moreover, the introduction of chemical functionalities further widens the application scope of such materials.

## 1. Introduction

Nowadays, biodegradable and renewable materials play an important role in research and industry. The most abundant biopolymer on Earth, cellulose, has been used by our society for centuries and is indispensable in today’s economies. In cellulose-related research, the interest in nanostructured celluloses is steadily rising. This group of celluloses includes cellulose nanocrystals (CNC), bacterial cellulose (BC), and cellulose nanofibrils (CNF) [1,2]. Significantly, CNF and BC, as high aspect ratio nanocelluloses, are frequently used to reinforce composite materials, e.g., 3D silica composites [3,4]. Three-dimensional silica composites based on alkoxysilanes are usually prepared from tetraethyl orthosilicate (TEOS) via sol–gel chemistry in a two-step gelation process: (A) TEOS is hydrolyzed in the presence of an acid catalyst, an organic solvent, and water (usually EtOH/water) [5], yielding a low-viscous precursor consisting of partially or completely hydrolyzed TEOS monomers and oligomers, i.e., the silica sol [6,7,8]. (B) The gelation of this precursor is then induced under basic conditions, e.g., through the addition of a base catalyst, which causes condensation reactions and the formation of a siloxane polymer network, i.e., the silica gel [5]. The preparation of silica gels via sol–gel chemistry has many advantages: it is a robust and well-investigated method, and functional gels can be easily obtained using special silanes (e.g., 3-aminopropyl triethoxysilane and 3-mercaptopropyl trimethoxysilane) to tune the properties of the end products [9]. Moreover, a variation of the synthesis and drying parameters gives access to a variety of materials, such as fine powders, ceramics, glasses, spherical particles, and nanocomposites gels [5]. Pure silica aerogels feature large specific surface areas (SSAs) ranging from 500 to 1000 m^2^/g [10]. However, the application of pure silica gels is limited due to their fragility [11]. Their mechanical properties can be improved through the introduction of polymeric materials [12] or (nano)celluloses [3,4,13,14]. CNF-based and BC-based silica composites were produced through the immersion of porous cellulose aerogels into the silica sol followed by subsequent gelation induced by the addition of ammonia acting as a base catalyst. The main drawbacks of these methods are: (1) The required drying step to produce a cellulose aerogel precursor is time-consuming and limits the processability into arbitrary 3D shapes. (2) Most methods are based on organic solvents, whose use should be avoided or at least reduced for sustainability reasons. (3) Finally, CNF–silica nanocomposites are handled as dry aerogels, and the direct preparation and utilization of CNF–silica hydrogels remains largely untapped.

The focus of this work lies in the fabrication of shapable, mechanically robust CNF–silica hydrogels, using an environmentally benign and solvent-free process. We studied first the influence of the solvent on the properties of the gels and tested methylcellulose (MC) and starch as an additive to increase the compressive strength of CNF–silica gels. Finally, we demonstrate the versatility of our approach through the in situ introduction of functional silanes bearing thiol or carboxylate groups and examined the functionalized gels for utilization as adsorbers for heavy metals and dyes.

## 2. Materials and Methods

### 2.1. Materials

Cellulose nanofibers were produced from never-dried bleached birch pulp through disintegration in an M110P microfluidizer (Microfluidics Crop., Newton, MA, USA). The pulp was fibrillated by 12 passes through the microfluidizer. Methylcellulose (product number: M0262, 413 cps, 30.3% methoxyl content) and potato starch (product number: S4251, 25.9 µm granule size, 31% amylose, DP_w_ of amylopectin subfraction: 35) [15] were purchased from Sigma-Aldrich (Sigma-Aldrich Chemie GmbH, Munich, Germany). *N*-[3-Trimethoxysilyl)propyl] ethylenediamine triacetic acid trisodium salt was purchased from abcr (abcr GmbH, Karlsruhe, Germany) as a 45% aqueous solution. All other used chemicals were, if not otherwise noted, purchased from Sigma-Aldrich with a purity of at least 99%.

### 2.2. Methods

#### 2.2.1. Preparation of CNF–Silica Hydrogels

Here, 12 mL of 1 wt% CNF dispersion (0.12 g dry cellulose), TEOS (98% purity, 2 mL, 1.9 g, 9.1 mmol), and aqueous HCl (0.16 mL, 0.29 M, 46 µmol HCl) were stirred overnight to hydrolyze TEOS to yield the CNF–silica sol. Then, 0.85 mL of 0.1 mol/L NH_3_ (85 µmol) were added and mixed quickly to start the condensation. The solution was filled into syringes to obtain hydrogels with cylindric shape (the shape can be freely adjusted by using different molds or other processing techniques). After complete gelation, the silica nanocomposites were aged in water at 50 °C for at least 10 h to stiffen the silica gel network. The gels were stored in DI water in the fridge.

#### 2.2.2. Preparation of CNF–MC–Silica Hydrogels

Here, 7 mL of 1.7 wt% CNF dispersion (0.12 g of dry cellulose), 3.5 mL of water, methylcellulose (1.5 mL, 2 wt%, 0.03 g), 2 TEOS (2 mL, 1.9 g, 9.1 mmol), and HCl (0.16 mL, 0.29 M, 46 µmol HCl) were stirred overnight to hydrolyze TEOS. Then, 0.85 mL of 0.1 mol/L NH_3_ (85 µmol) was added and mixed quickly to start the condensation. The solution was filled into syringes to obtain hydrogels with cylindrical shapes. Finally, CNF–MC–Silica hydrogels were aged in water at 50 °C for at least 10 h to stiffen the network.

#### 2.2.3. Preparation of CNF–Starch–Silica Hydrogels

TEOS (2 mL; 1.9 g, 9.1 mmol) was hydrolyzed in the presence of 7 mL of 1.7 wt% CNF (0.12 g dry cellulose), HCl (0.16 mL, 0.29 M, 46 µmol HCl), and 4 mL of water. Then, a 3 wt% dispersion of starch was prepared by adding the starch into cold deionized water and heating under agitation to 80 °C to allow gelatinization. Then, 1 mL of this mixture was cooled down to 60 °C at room temperature, and then directly added into the CNF–silica sol. Then, 0.85 mL of 0.1 mol/L NH_3_ (85 µmol) was added and mixed quickly to start the condensation. The solution was filled into syringes to obtain hydrogels of a cylindrical shape. Finally, CNF–Starch–Silica hydrogels were aged in water at 50 °C for at least 10 h to stiffen the network.

#### 2.2.4. Preparation of Thiol-Functionalized CNF–Starch–Silica Hydrogels

The respective hydrogels were prepared according to the above procedure for the preparation of CNF–silica hydrogels, using the same quantities of reactants. After acidic hydrolysis, (3-mercaptopropyl)trimethoxysilane (MTMS) (95% purity, 0.13 mL, 0.14 g, 0.7 mmol) was added into the prepared CNF–silica sol (before the addition of ammonia). Upon addition of the mercapto silane, the sample was protected from light with aluminum foil. Then, 0.85 mL of 0.1 mol/L NH_3_ (85 µmol) was added, and the sample was transferred into a mold. After complete gelation, the silica nanocomposites were aged in water at 50 °C for at least 10 h to stiffen the network. The gels were stored in DI water in the fridge.

#### 2.2.5. Preparation of Carboxylate-Functionalized CNF–Starch–Silica Hydrogels

The respective hydrogels were prepared according to the above procedure for the preparation of CNF–silica hydrogels, using the same quantities and procedure. Directly after the addition of ammonia, the functional silane, *N*-[3-trimethoxysilyl)propyl] thylenedi-amine triacetic acid trisodium salt (0.13 mL, 0.16 g, 0.3 mmol), was added to the CNF–silica mixture. The sample was directly transferred into a mold. After complete gelation, the silica nanocomposites were aged in water at 50 °C for at least 10 h to stiffen the network. The gels were stored in DI water in the fridge.

#### 2.2.6. Compression Tests

The mechanical properties of the composites were measured on a universal testing machine Zwick/Roell Z020 (Ulm, Germany). Compression tests were performed in a wet state, with a 500 N load cell. The strain rate was set to 2.4 mm/min and samples were compressed to 30%. The compressive strength was defined as the maximum stress in the performed strain range. Measurements were performed in triplicate and compared with Student’s *t*-test (unpaired, *n* = 3).

#### 2.2.7. Solvent Exchange to *Tert*-BuOH and Freeze-Drying

Samples were freeze-dried from the respective *t*BuOH gel to avoid ice-templating effects and to preserve the gels’ nanostructure upon the freeze-drying process [16]. The samples were solvent-exchanged first with 1:1 *t*BuOH/water, second with 8:2 *t*BuOH/water, and finally with pure *t*BuOH (each solvent exchange step was conducted with solvent amounts of approx. ten times the respective sample volume and equilibrated for 24 h on a shaker). All samples were freeze-dried in a Christ Beta 1–8 LD Plus freeze-dryer. After the last step of solvent exchange with 100% *t*BuOH, the samples were taken out of the system and frozen at −80 °C in a glass vial. The frozen samples were quickly transferred into the lyophilizer. All samples were freeze-dried for at least 24 h and stored afterward in an airtight container.

#### 2.2.8. Specific Surface Area

The 3D composites for the BET measurement were prepared by cutting the dried samples into small pieces, pre-drying them at 60 °C for at least 24 h, and storing them in an airtight beaker with drying beads to keep the samples dry. The samples were degassed in a FlowPrep 060 (Mircomeritics, Norcross, GA, USA) at 80 °C under N_2_ flow for at least 6 h. Afterward, the measurement was performed on a Micromeritics TriStarII instrument.

#### 2.2.9. Porosity of Samples

The porosity of the 3D composites was calculated according to Equation (1).

(1)
$$\Phi =\left(1-\frac{\rho }{w_{s}*\rho_{s}+w_{c}*\rho_{c}}\right)*100$$

where 
ρ
 is the bulk density of the sample (measured by gravimetric means from dried samples with defined volume), and the porosity was calculated in percent. 
$\rho_{s}$
 is the density of silica, at 2.19 g/cm^3^ [17], and 
$\rho_{c}$
 is the density of cellulose, at 1.59 g/cm^3^ [18]. 
$w_{s}$
 and 
$w_{c}$
 are the mass fraction of silica and cellulose, respectively, calculated using Equation (2).

(2)
$$w_{s}\left[\%\right]=\left(\frac{\rho -\rho_{0}}{\rho }\right)*100$$

where 
$\rho_{0}$
 stands for the cellulose bulk density in the sample, calculated from native CNF or BC gels cryogels. 
$\rho_{0}$
 = 0.010 g cm^−3^ in case of CNF and 0.007 g cm^−3^ in case of BC.

#### 2.2.10. Scanning Electron Microscopy (SEM)

Micrographs of freeze-dried samples were measured on a Zeiss Supra 55 VP. Before the measurement, the samples were sputtered with a 10 nm-thick gold layer (Leica EM SCD050, Wetzlar, Germany).

#### 2.2.11. Detection of Thiol Groups

The number of thiol groups in the composite sample obtained upon the addition of MTMS was quantified using Ellman’s test according to the instructions of Thermo Scientific [19]. Ellman’s test was performed in a reaction buffer containing 0.1 M sodium phosphate solution and 1 mM of ethylenediaminetetraacetic acid at a pH of 8. Ellman’s reagent solution was prepared as follows: 4 mg Ellman’s reagent (10 µmol), i.e., 5,5′-dithio-bis-(2-nitrobenzoic acid), was dissolved in 1 mL of the reaction buffer. A solution of 2.5 mL of reaction buffer and 50 µL of Ellman’s reagent solution were prepared for each sample (including blank and positive control). For the blank sample, 250 µL of reaction buffer was added to the prepared solution. For the positive control, 4 µmol of MTMS and 246 µL of reaction buffer were added. To measure the number of thiols in the hydrogels, 10–40 mg hydrogel of each sample was added to the prepared solutions and the volume was adjusted to 2.8 mL with reaction buffer. The total thiol content was detected by UV-VIS spectroscopy (Perkin Elmer Lambda 35, Waltham, MA, USA) at a wavelength of 412 nm.

#### 2.2.12. Adsorption of Copper(II) Sulfate

A copper(II) sulfate solution with a concentration of 500 ppm was prepared using copper (II) sulfate pentahydrate. Here, 8.5 mL of this solution was added to the hydrogel sample (0.9–1.2 g of hydrogel weight with approx. 6% solid content). The performance of the carboxylate-functionalized CNF–silica composite was compared to native CNF–silica composite and a blank sample. The amount of adsorbed copper(II) was measured by monitoring the residual copper(II) concentration in the solution after 8 days of equilibration, which was determined by UV-VIS spectroscopy at 812 nm. The UV-VIS measurements for this sample were carried out without further dilution.

## 3. Results and Discussion

All composites were prepared via sol–gel chemistry starting from tetraethyl orthosilicate (TEOS). In combination with nanocellulose, mechanically robust hydrogels were obtained. The first experiments aimed at studying the effect of EtOH/water ratio on the mechanical properties and specific surface area (SSA) of cellulose/silica gels. These studies were conducted with bacterial cellulose as a model material since it naturally provided a stiff, pre-shaped and mechanically robust gel network. For follow-up experiments, BC was replaced with CNF, a highly viscous, shapable suspension, which allows an easy adjustment of the final shapes of the gels (Figure 1).

We compared the gelation in an EtOH/water ratio of 5:1 (*v*:*v*) with our completely water-based and solvent-free method. The prepared hydrogels were tested with uniaxial compression tests to evaluate their mechanical resistance (Appendix A and Appendix A). In addition, the properties of respective dry gels were compared to study their specific surface area, density, and porosity (Appendix A). Cryogels [20] were prepared via freeze-drying after solvent exchange to *t*BuOH [21]. This gives highly porous materials, which are very similar to conventional aerogels prepared via supercritical CO_2_ (scCO_2_) drying. We selected this approach, since it is more frequently used and easier to perform than drying with scCO_2_, and the obtained composite cryogels featured comparable and even slightly higher specific surface area values than supercritical CO_2_ dried samples (Appendix A). Silica hydrogels—prepared according to our organic, solvent-free method—had significantly higher compressive strength values of 22 kPa in comparison to the 13 kPa of samples obtained using the EtOH/water mixture (Appendix A). The higher compressive strength can be explained by the higher density of the sample produced in the organic, solvent-free process. The main reasons for this effect are most probably the higher rates of the condensation reactions for increasing water content [22], which lead to a larger size of silica particles deposited on the BC structure as shown by scanning electron microscopy (Appendix A). Similar observations have been also made in TEOS model systems, showing that the aggregate/particle size grows with increasing water content during TEOS gelation [23]. The larger particle size in the organic, solvent-free process explains its smaller specific surface area (759 m^2^ g^−1^ vs. 897 m^2^ g^−1^) and lower porosity (Appendix A). These values are in the range of reported properties of BC silica composites [3]. In comparison to previous approaches [3,4,13], no drying step of nanocellulose is required, and we were able to directly use wet nanocellulose gels. To generate gels with high mechanical strength, we used our organic, solvent-free approach in the subsequent CNF composite preparation.

We replaced BC with CNF to further increase the versatility of our method (Figure 1) and allow the production of silica composite gels of arbitrary shape (Figure 1D,E). In this process, TEOS was hydrolyzed in a CNF suspension in the presence of HCl as an acidic catalyst for the preparation of CNF–silica gels or CNF in combination with soluble polysaccharides (PS; methylcellulose (MC), starch) to obtain CNF–PS–Silica gels. After complete hydrolysis (usually after stirring overnight), the respective sol (Figure 1B) can be transferred into a mold or processed with an extrusion technique, such as 3D printing, for shaping purposes. The addition of ammonia finally triggered the gelation of the silica gels, and the gel network was densified and stiffened (causing gel shrinkage) through curing/aging at 50 °C in water (Figure 1C,D).

The mechanical properties of the gels are compared to a BC–Silica gel (Figure 2 and Table 1). Because of the already-strong gel network of native BC, the prepared BC–Silica gels featured higher compressive strength than the CNF–Silica gel (22 kPa vs. 14 kPa). Since our aim was the preparation of mechanically robust composite gels, we studied the addition of two soluble polysaccharides—starch or MC—to the gel network to strengthen it. The addition of MC slightly raised the compressive strength by approx. 10%, whereas the starch addition almost doubled the compressive strength from 14 kPa (CNF–Silica gel) to 26 kPa (CNF–Starch–Silica gel).

Respective cryogels were prepared by freeze-drying, after solvent exchange to *t*BuOH, for analytical purposes. Native CNF cryogels featured a specific surface area of 135 m^2^ g^−1^ and a porosity of 99% (Table 1), which is comparable to other values in the literature [24,25]. CNF–Silica gels had a much higher specific surface area of 603 m^2^ g^−1^ and density due to the incorporation of the silica network (Figure 3B). MC and starch increased the densities and enlarged the SSA. The increase in SSA was especially evident in the case of CNF–MC–Silica, which increased from 603 m^2^/g (CNF–Silica) to 740 m^2^/g. As shown in Figure 3, the native CNF network is covered with silica particles in the composite samples. The CNF network structure is visible in the case of CNF–Silica and CNF–MC–Silica cryogels (Figure 3B,C), in which individual fibrils are covered with silica particles. In contrast to that, CNF–Starch–Silica featured a very different cauliflower-like nanostructure (Figure 3D), which covered nearly completely the fibrillar skeleton of CNF. The drastic change in nanostructure in the case of CNF–Starch–Silica cryogels is in line with the higher density, lower porosity, and increased compressive strength (Table 1).

As summarized in Figure 1, our proposed method is straightforward, organic, solvent-free, and enables the preparation of moldable CNF–silica gels of high compressive strength. The prepared CNF–silica gels also featured large specific surface areas in the range of 603–740 m^2^ g^−1^.

Besides TEOS, alkoxysilanes are commercially available with a wide range of additional chemical functionalities, and those functional silanes were shown, e.g., to be suitable for the functionalization of pristine CNFs [26,27,28,29]. The addition of chemical functional groups onto the surface of the prepared composites would further increase their application range and versatility. We tested the in situ modification of the CNF–Silica composites with alkoxysilanes bearing thiol (Figure 4A1) and carboxylate groups (Figure 4A2).

Functional CNF–Silica gels were prepared with 0.3–0.7 mmol (3–8% functionalization degree based on the total molar amount of TEOS) of the respective alkoxysilane (Figure 4A1,A2), 3-mercaptopropyl trimethoxysilane (MTMS), or *N*-[3-trimethoxysilyl)propyl] ethylenediamine triacetic acid trisodium salt (3CTMS). In the following section, we focus on the results of functional CNF–Silica hydrogels, this protocol can be also applied to the mechanically robust polysaccharide-reinforced composites, CNF–MC–Silica, and CNF–Starch–Silica. Since the sol–gel process of silica gels is largely dependent on the pH, the acidity/basicity of the functional silane must be considered, e.g., an uncontrolled change of pH could prevent gelation or cause uncontrolled gelation and the formation of isolated gelled lumps (this has to be taken into account if the compatibility of our method to other functional silanes is tested). The functional silanes were first added directly into the silica sol after the TEOS hydrolysis step to avoid influences on the hydrolysis and early reaction of TEOS and the functional silane. MTMS did not influence the pH and could be added into the silica sol without any modification. Upon addition of MTMS, the gelation was triggered, as in our standard protocol with ammonia, where we noted that no shrinkage occurred during the aging step. A similar effect was observed in the literature in the case of hydrophobic silanes [30]. Due to the hydrophilicity of the thiol groups and their susceptibility to oxidation, we assume that disulfide crosslinks are partially formed preventing shrinkage. We proved the successful introduction of thiol groups into the composite with Ellman’s test, which allows the quantitative detection of thiols [31]. The incorporated amount of thiols was determined to be 1.3 µmol g^−1^ gel. Due to the high reactivity of these groups, they can be post-modified with high efficiency using, e.g., thiol–epoxy [32] or thiol–ene [33] and thiol–Michael [34] click chemistry.

Due to the basicity of 3CTMS (carboxylate form), the first tests were unsuccessful and the addition of 3CTMS into CNF–Silica sol neutralized the medium, causing rapid, uncontrolled gelation. To avoid this, we adapted the method and added ammonia directly before 3CTMS. Thereby, the condensation rate was reduced, and we were able to produce stable CNF–Silica gels functionalized with carboxylate groups. We tested these carboxylated silica hydrogels in two different adsorber applications: the adsorption of copper(II) cations and the adsorption of the cationic dye methylene blue. All adsorption experiments were conducted for 8 days and the amount of adsorbed Cu(II) was determined with photometry. The carboxylated CNF–Silica gel adsorbed up 0.58 mg Cu(II)/g of hydrogel, whereas the native CNF–Silica gel was able to bind only 0.04 mg/g (Figure 4B). This demonstrates a higher adsorption capacity of the functional gel and proves as well the successful incorporation of the carboxylate groups. Furthermore, we tested the adsorption of methylene blue (a 20 ppm test solution) as part of our qualitative experiments. Additionally, in this case the dye adsorption capacity of the CNF–silica gel modified with 3CTMS was significantly higher than that of the CNF–silica gel counterpart (Figure 4C).

## 4. Conclusions

In this contribution, we demonstrated that CNF–Silica hydrogels can be produced straightforwardly with an organic, solvent-free process. In our process, no prior drying step is required, and the CNF can be directly dispersed in the silica sol. Gelation is finally triggered through the addition of ammonia, and the samples can be easily shaped through molding or other processing techniques suitable for CNFs. We further increased the compressive strength of those hydrogels through the addition of soluble polysaccharides. Significantly, the addition of starch was very promising and significantly increased the compressive strength (26 kPa) while preserving the specific surface area of the gels. Depending on the process, the specific surface area of the CNF–silica samples can be tuned from 603 m^2^ g^−1^ to 740 m^2^ g^−1^. Finally, we showed that alkoxysilanes bearing thiol and carboxylate groups can be incorporated to introduce functional groups onto the composite gels.

## Figures and Tables

**Figure 1 nanomaterials-12-00895-f001:**
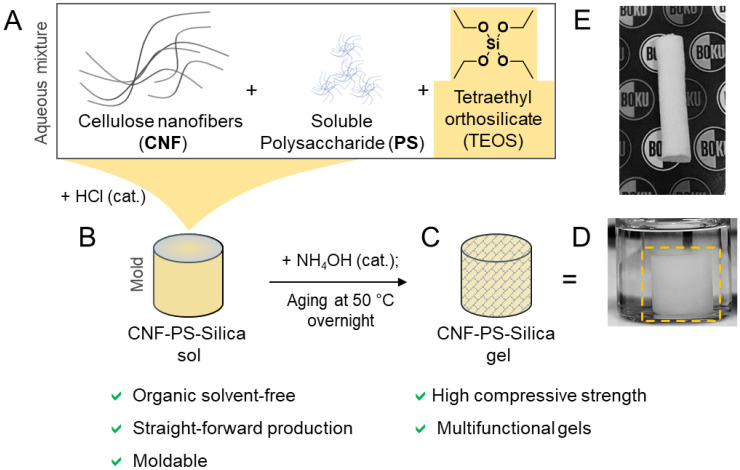
Preparation of silica composite gels reinforced with cellulose nanofibers (CNF) and a soluble polysaccharide (PS), starch, or methylcellulose. (**A**) The individual components are mixed in the presence of catalytic amounts of HCl to catalyze the hydrolysis of tetraethyl orthosilicate (TEOS). (**B**) Subsequently, ammonia is added to increase the pH and trigger the condensation and, thereby, the gelation of the sample. This was followed by the aging of the samples at 50 °C to stiffen the gel network and obtain the final CNF–PS–Silica gel (**C**). The respective hydrogel (**D**) was dried by freeze-drying after solvent exchange to obtain highly porous CNF–PS–silica cryogels (**E**). The shape of the gel can be controlled through molding or 3D printing of the CNF–PS–silica sol.

**Figure 2 nanomaterials-12-00895-f002:**
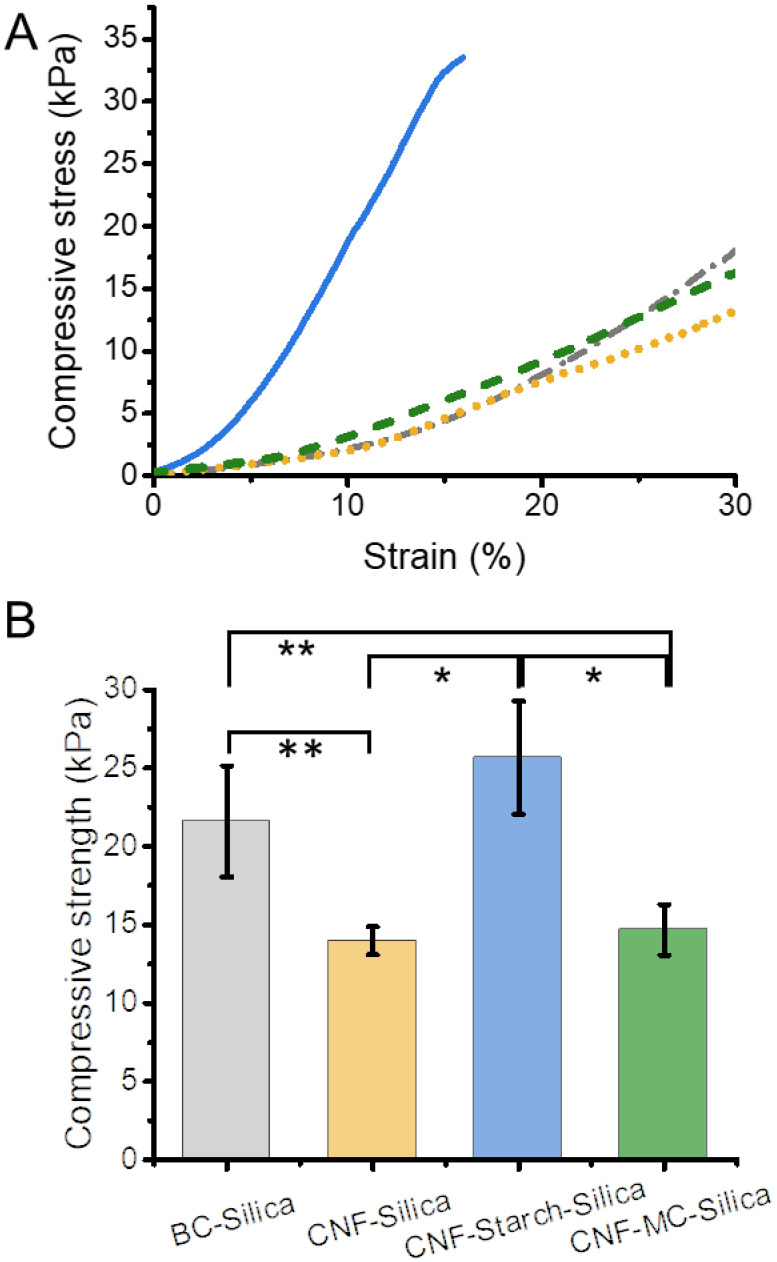
Mechanical properties of the prepared CNF–silica hydrogels and influence of the addition of the soluble polysaccharides—starch or methylcellulose (MC)—on the compression behavior. (**A**) Compression tests of the CNF–Silica hydrogels up to 30% strain (CNF–silica: dotted yellow line, CNF–MC–Silica: green dashed line, and CNF–Starch–Silica: blue solid line) in comparison to bacterial cellulose (BC) silica gel (dash-dotted gray line). (**B**) Comparison of the average compressive strengths and their standard deviation of the samples. Highlighted differences are statistically significant (* *p* < 0.04, ** *p* < 0.04, *n* = 3).

**Figure 3 nanomaterials-12-00895-f003:**
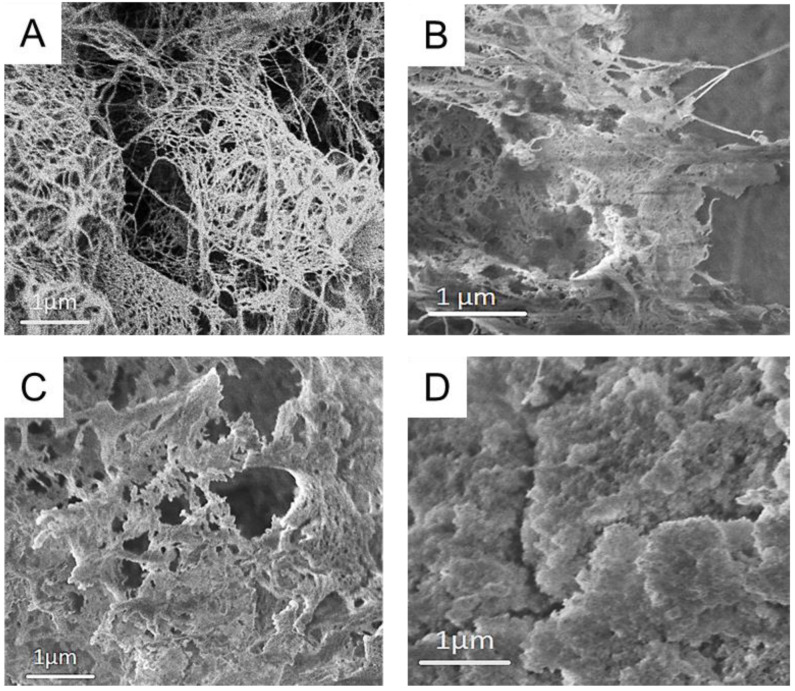
Scanning electron micrographs of CNF (**A**), CNF–Silica (**B**), CNF–Methylcellulose–Silica (**C**), and CNF–Starch–Silica (**D**) cryogels.

**Figure 4 nanomaterials-12-00895-f004:**
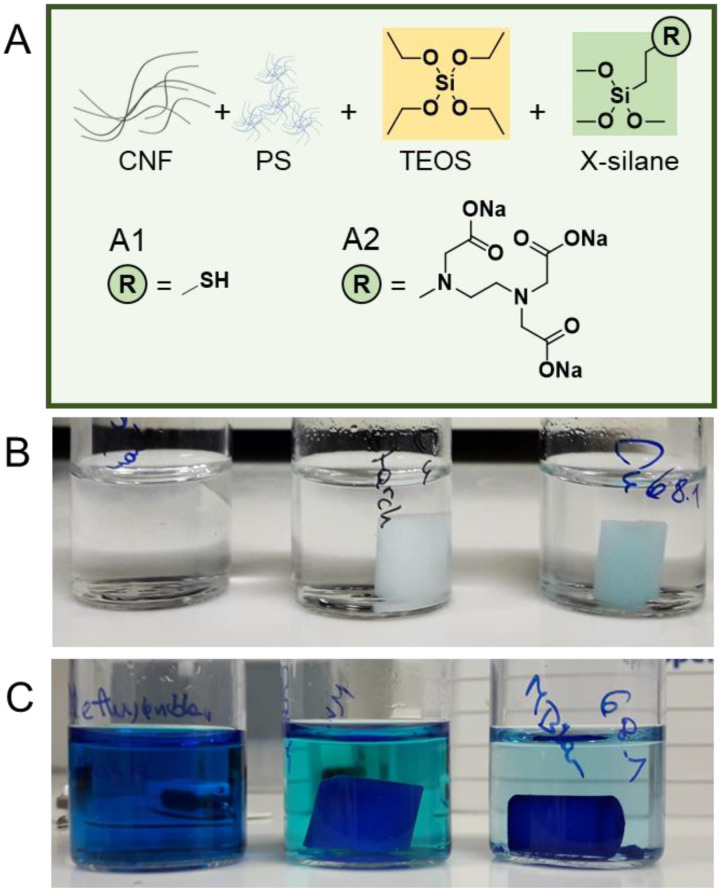
(**A**) Preparation of functional CNF–Silica hydrogels through the addition of either 3-mercaptopropyl trimethoxysilane (**A1**) or *N*-[3-trimethoxysilyl)propyl] ethylenediamine triacetic acid trisodium salt (**A2**) to introduce either thiol or carboxylate structures onto the hydrogel. Carboxylated CNF–Silica gels were tested as adsorbers for Cu(II) ions (**B**) and methylene blue (**C**). Vials in (**B**,**C**) contain (left to right): solutions before the addition of gels, non-functional CNF–Silica gel, and carboxylated CNF–Silica gel.

**Table 1 nanomaterials-12-00895-t001:** Properties of CNF–silica gels and the influence of the addition of starch or methylcellulose (MC). The standard deviation of the average compressive strengths is reported (*n* = 3).

Samples	Density (g cm^−3^)	Specific Surface Area (m^2^ g^−1^)	Porosity (%)	Compressive Strength * (kPa)
CNF–Silica	0.061	603	97.1	14 ± 1
CNF–MC–Silica	0.062	740	97.1	15 ± 2
CNF–Starch–Silica	0.065	625	96.9	26 ± 4
CNF	0.010	135	99.0	-

* Measurements were conducted from 0 to 30% strain, and the highest compressive stress value in this range was defined as compressive strength.

## Data Availability

The data that support the findings of this study are available on request from the corresponding author, M.B.

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
