# Peer review of "Facile Preparation of Mechanically Robust and Functional Silica/Cellulose Nanofiber Gels Reinforced with Soluble Polysaccharides"

_nanomaterials, 2022, doi:10.3390/nano12060895_

Round 1
Reviewer 1 Report
This manuscript is well written and very interesting. The only thing missing in this research work is the characterization of raw materials used in the formulation of hydrogels :
- for methylcellulose, viscosity is indicated but neither polymerization degree nor degree of methylation
- for starch: even if authors indicated that they used potato starch, it would be appreciated if they indicate DP and amylopectin ration.
- for soluble polysaccharide : DP (or MW)
Author Response
Dear Reviewer, thank you for your positive feedback and suggestions. We addressed all comments and added further information on the used biopolymers (article number and properties). See further details in the manuscript.
Reviewer 2 Report
The submitted manuscript is of high interest for the readers of nanomaterials and is therefore suitable for publication. Overall, I have no major concerns regarding the content and the presentation of the results.
I have however a problem with the use of certain terms. The main is the use of the word “sustainable”. I do not find this product sustainable, it is cellulose-based, yes, but then the addition of TEOS and the use of certain products during the elaboration of the gels are not at all sustainable. I invite the authors to consider this issue, as the concept of a sustainable product involves the whole life cycle, cradle to cradle. So please consider using a more appropriate word as can be biobased, cellulose-based, plant-based, or similar. The second is the term “aerogels”, I consider that the authors have very proficient gels, but they are not aerogels, they are rather cryogels, please refer to the work: 10.1007/s10570-016-0960-8to have a broader idea of gel definitions.
The kinetics of the distribution of silica particles (lines 217-222 can be explored further as this is a very interesting phenomenon, the authors can make a deeper relation between the uniformity and specific properties of the gels, moreover, the particle size and particle distribution within the gel can be explained in more detail, as well as the reasons for this.
Please provide statistical comparisons for the means in the mechanical tests (perhaps a Tukey or Bonferoni test) to have stronger bases to compare these results.
Author Response
- The submitted manuscript is of high interest for the readers of nanomaterials and is therefore suitable for publication. Overall, I have no major concerns regarding the content and the presentation of results.
Thank you so much for the very positive feedback. Please note that we revised the manuscript based on the comments below (highlighted in yellow color in the text).
- I have however a problem with the use of certain terms. The main is the use of the word “sustainable”. I do not find this product sustainable, it is cellulose-based, yes, but then the addition of TEOS and the use of certain products during the elaboration of the gels are not at all sustainable. I invite the authors to consider this issue, as the concept of a sustainable product involves the whole life cycle, cradle to cradle. So please consider using a more appropriate word as can be biobased, cellulose-based, plant-based, or similar. The second is the term “aerogels”, I consider that the authors have very proficient gels, but they are not aerogels, they are rather cryogels, please refer to the work: 10.1007/s10570-016-0960-8to have a broader idea of gel definitions.
Thank you for this remark. We removed the term sustainable from the title. In addition, we exchanged the term aerogels with cryogels to be more specific. We would like to highlight that the chosen freeze-drying method used with prior solvent exchange to tert-BuOH is very different from conventional freeze-drying; yields aerogel-like materials. Interestingly, our SSA values were very similar or even slightly higher than reference samples obtained via scCO2 drying (see SI for further information). We also added the recommended citation, which we believe is very relevant to our work.
- The kinetics of the distribution of silica particles (lines 217-222 can be explored further as this is a very interesting phenomenon, the authors can make a deeper relation between the uniformity and specific properties of the gels, moreover, the particle size and particle distribution within the gel can be explained in more detail, as well as the reasons for this.
This is indeed very interesting, and we are planning to further analyses to get a better understanding of the gelation in such conditions. We added as well further literature and a short discussion to this passage. The increase in particle size and kinetics has been shown also in other works in the literature.
- Please provide statistical comparisons for the means in the mechanical tests (perhaps a Tukey or Bonferoni test) to have stronger bases to compare these results.
This is a very relevant suggestion, thank you. We added statistic analyses (t-test), see Figure 2B, and added as well further related information.
Reviewer 3 Report
This is a well-thought-out approach to prepare high-surface-area CNF/silica composite gel with excellent mechanical properties but still with mouldable workability. The characterizations involved in the materials are in-depth and very elaborately presented. The versatility in application potential is supported with the ease to functionalize the gel with the integration of alkoxysilanes as functional groups, which is also very convenient as it can be performed in a one-pot synthesis of the composite gel.
Some discussions to the authors:
1) The surface properties of CNF used in the present study in low surface charge, as it is indicated in the method section. What would be the role of surface charge on nanofiber in the process of synthesis of such composite gels, such as negatively surface charge that can be commonly introduced by TEMP-oxidization?
2) The application of such a composite gel system can be of high value as an interfacing matrix in medical and therapeutic devices, as the silica gel itself is a very successful candidate in various biomedical and pharmaceutical applications, thanks to its biodegradability.
Author Response
Thank you so much for your very encouraging feedback.
- The surface properties of CNF used in the present study in low surface charge, as it is indicated in the method section. What would be the role of surface charge on nanofiber in the process of synthesis of such composite gels, such as negatively surface charge that can be commonly introduced by TEMP-oxidization?
In our work, we used a native CNF sample, which has rather a low charge density in comparison to chemically modified CNFs, such as TEMPO-CNFs. Your comment is very relevant and is something we will consider and subsequent studies. We believe that also the type of CNF will have significant influences on the silica gel morphology. We tested during the preparation of this study two different mechanically fibrillated samples (coarse vs. fine) and realized that the size of the fibrils has a major influence on the properties of the prepared gels. Based on our results, we used the fine CNF sample for the preparation of hydrogels. Consequently, the CNF charge might be also very relevant and is something we aim to address in follow-up work.
- The application of such a composite gel system can be of high value as an interfacing matrix in medical and therapeutic devices, as the silica gel itself is a very successful candidate in various biomedical and pharmaceutical applications, thanks to its biodegradability.
Our work is a proof-of-principle demonstrating that functional silica hydrogels can be prepared in a straightforward as well as organic solvent-free manner, your suggestion for possible applications is very appreciated. We will consider this in our follow-up studies and are hopeful that this work can stimulate collaborations, as well as further advancement in the field.